# Adsorption of Phenazines Produced by *Pseudomonas aeruginosa* Using AST-120 Decreases Pyocyanin-Associated Cytotoxicity

**DOI:** 10.3390/antibiotics10040434

**Published:** 2021-04-13

**Authors:** Hidetada Hirakawa, Ayako Takita, Motoyuki Uchida, Yuka Kaneko, Yuto Kakishima, Koichi Tanimoto, Wataru Kamitani, Haruyoshi Tomita

**Affiliations:** 1Department of Bacteriology, Graduate School of Medicine, Gunma University, Maebashi, Gunma 371-8511, Japan; takita626@gunma-u.ac.jp (A.T.); m1800025@gunma-u.ac.jp (Y.K.); m1800021@gunma-u.ac.jp (Y.K.); tomitaha@gunma-u.ac.jp (H.T.); 2R&D Strategy & Planning Department, Kureha Corporation, 16 Ochiai, Iwaki, Fukushima 974-8686, Japan; uchida@kureha.co.jp; 3Laboratory of Bacterial Drug Resistance, Graduate School of Medicine, Gunma University, Maebashi, Gunma 371-8511, Japan; tanimoto@gunma-u.ac.jp; 4Department of Infectious Diseases and Host Defense, Graduate School of Medicine, Gunma University, Maebashi, Gunma 371-8511, Japan; wakamita@gunma-u.ac.jp

**Keywords:** antimicrobial resistance (AMR), multi-drug resistance (MDR), quorum sensing, virulence, bacterial pathogenesis, molecular genetics, signal transduction, infection control

## Abstract

AST-120 (Kremezin) is used to treat progressive chronic kidney disease by adsorbing uremic toxin precursors produced by the gut microbiota, such as indole and phenols. Previously, we found that AST-120 decreased drug tolerance and virulence in *Escherichia coli* by adsorbing indole. Here, we show that AST-120 adsorbs phenazine compounds, such as pyocyanin, produced by *Pseudomonas aeruginosa* including multidrug-resistant *P. aeruginosa* strains, and suppresses pyocyanin-associated toxicity in A-549 (alveolar adenocarcinoma) and Caco-2 (colon adenocarcinoma) cells. Addition of fosfomycin, colistin and amikacin, which are often used to treat *P. aeruginosa*, inhibited the bacterial growth, regardless of the presence or absence of AST-120. These results suggest a further benefit of AST-120 that supports anti-Pseudomonas chemotherapy in addition to that of *E. coli* and propose a novel method to treat *P. aeruginosa* infection.

## 1. Introduction

*Pseudomonas aeruginosa* is a well-studied opportunistic pathogen. This bacteria is frequently found in patients suffering from chronic respiratory infections, such as cystic fibrosis (CF) and diffuse panbronchiolitis, and it eventually causes respiratory failure in patients [1,2]. Serious infections also occur in compromised hosts when *P. aeruginosa* enters the bloodstream from multiple possible sites, such as burn wound sites, the urinary tract and the intestinal tract, resulting in fatal sepsis [3,4,5]. Treatment of *P. aeruginosa* infections is generally difficult because of its innate resistance and tolerance to a broad range of antimicrobial agents [6,7]. In addition, advanced multidrug-resistant strains have often been isolated in clinical settings [8,9].

*P. aeruginosa* produces a large number of molecules that contribute to its virulence. Phenazines are one such group of molecules, of which pyocyanin is the most well characterized phenazine molecule. This molecule is synthesized from the shikimic acid substrate [10]. Some studies have suggested the crucial roles of pyocyanin in its toxicity to host cells. This molecule causes oxidative stress in epithelial cells and disrupts Ca^2+^ homeostasis associated with the V-ATPase activity while also decreasing the immune response by inducing apoptosis in neutrophils and suppressing the secretion of immunoglobulins by B-lymphocytes and the proliferation of T-lymphocytes [11,12,13,14]. In addition, phenazine/pyocyanin-deficient *P. aeruginosa* mutants exhibit a low fatality rate in *Caenorhabditis elegans*, *Drosophila* and burned mice and a reduced infection rate in the lungs of mice [15,16,17,18].

AST-120 (Kremezin) is a carbonaceous adsorbent, used as an oral drug to treat progressive chronic kidney disease (CKD) by delaying the disease’s progression and the initiation of dialysis. In enteric sites, AST-120 adsorbs uremic toxin precursors, including indole produced by enteric bacteria, and reduces the production of uremic toxins, such as indoxyl sulfate [19]. In addition to its efficacy in treating CKD, we previously found that indole induced drug tolerance and virulence in *Escherichia coli* including enterohaemorrhagic *E. coli*, and AST-120 eliminated these effects by adsorbing indole. Thus, AST-120 reduces drug tolerance and virulence in *E. coli* [20]. AST-120 can also adsorb some other small-sized molecules that have aromatic ring structures in addition to indole [10]. Here, we show evidence that AST-120 adsorbs phenazines, including pyocyanin produced by *P. aeruginosa* including clinical isolates that are resistant to antimicrobials, such as multidrug-resistance *P. aeruginosa* (MDRP), and reduces pyocyanin-associated toxicity to host epithelial cells.

## 2. Results

### 2.1. AST-120 Adsorbs Phenazines Produced by P. aeruginosa and Decreases the Extracellular Pyocyanin Levels

Pyocyanin is one of the phenazine molecules and it is produced from its phenazine precursors such as phenazine-1-carboxylic acid and 5-methylphenazine-1-carboxylic acid betaine (Figure 1). Since phenazine compounds, including pyocyanin, have tricyclic structures consisting of a benzene ring [10], we hypothesized that AST-120 might adsorb these molecules, decreasing the levels of pyocyanin in *P. aeruginosa* cultures. To test this hypothesis, we incubated a synthetic phenazine with AST-120 for 2 h in a bacteria-free solution and the level of phenazine in the supernatant was measured after the removal of AST-120. More than 90% of the phenazine was adsorbed when incubated with at least 10 mg of AST-120 (Figure 2A). The bacteria-free supernatant, filtrated from a 24 h culture of PAO1, was incubated with AST-120, and we measured the level of pyocyanin in the supernatant after the removal of AST-120 to assess the pyocyanin-adsorption ability of AST-120. We did not detect pyocyanin residues after incubation with at least 20 mg AST-120 (Figure 2B). We also measured the pyocyanin level in the culture supernatant of PAO1 when co-cultured with AST-120. The color of the PAO1 culture was clear when cultured in the presence of 30 mg AST-120 and the pyocyanin level was below the detection limit (Figure 2C,D). In contrast, the color of the bacterial culture was blue/green when grown in the absence of AST-120 (Figure 2C). PhzA1 and PhzB1 proteins are required for the synthesis of pyocyanin. Previously, the *phzB1* mutant was found to exhibit defective production of pyocyanin [21]. Here, we confirmed that the *phzA1/B1* mutant (designated ∆phzA1/B1) exhibited no pyocyanin production (Figure 2C). These observations suggest that AST-120 adsorbs phenazine products including pyocyanin produced by *P. aeruginosa*, and decreases the level of pyocyanin. In addition to PAO1 culture, we tested the effect of AST-120 on removing pyocyanin in some drug-resistant clinical isolates, including MDRP strains. We found that strains Ps.a-1200, Ps.a-1205 and RPs.a-947 highly produced pyocyanin, and the addition of 30 mg AST-120 reduced pyocyanin levels in their cultures, while RPs.a-914 exhibited a low pyocyanin production (Figure 2E). 

### 2.2. AST-120 Does Not Alter the Levels of LasI-LasR and RhlI-RhlR-Mediated Quorum Sensing

Pyocyanin production is induced by LasI-LasR and RhlI-RhlR quorum sensing (QS) systems [22]. We concluded that AST-120 does not alter the activities of these QS systems for the following reasons. LasI-LasR and RhlI-RhlR are known to elevate the activity of *lasI* and *rhlA* promoters, respectively. We measured the levels of *lasI* and *rhlA promoters* to determine LasI-LasR and RhlI-RhlR activities in PAO1 grown with and without 30 mg AST-120. There was no significant difference in β-galactosidase levels corresponding to the activity of *lasI* in the PAO1 strains grown with and without AST-120 while β-galactosidase level corresponding to the activity of *rhlA* in PAO1 grown with AST-120 was modestly lower than that grown without AST-120 (Figure 3). We also found that there was no significant difference in the growth of PAO1 when cultured in the presence and absence of AST-120 (data not shown). 

### 2.3. AST-120 Decreases Pyocyanin-Associated Toxicity of P. aeruginosa to Host Cells

Pyocyanin causes damage and death to host cells. Its function has been well defined in airway epithelial cells, such as A-549 (11). We tested whether removal of pyocyanin by AST-120 treatment reduces the toxicity to A-549 cells. The addition of a bacteria-free supernatant from a PAO1 culture killed approximately 80% of A-549 cells while 55% of the host cells survived when incubated with supernatant from PAO1 cultured with 30 mg AST-120 (Figure 4A). We also found that the cytotoxicity of supernatants from Ps.a-1200, Ps.a-1205 and RPs.a-946 cultures, but not RPs.a-914 (poor pyocyanin producer), was alleviated when the bacteria were grown in the presence of AST-120 (Figure 4A). When AST-120 is orally administrated, it may adsorb pyocyanin produced by *P. aeruginosa* in the gut and attenuate damage to intestinal cells. We determined the toxicity of supernatants from *P. aeruginosa* cultured with and without AST-120 to Caco-2 cells to test this hypothesis. The addition of supernatants from PAO1, Ps.a-1200, Ps.a-1205 and RPs.a-946 cultures killed more than 99.9% of Caco-2 cells while the cytotoxicity of culture supernatant from RPs.a-914 (the poor pyocyanin producer) was relatively modest (approximately 10% of the host cells survived). Similar to A-549 cells, the addition of AST-120 to the cultures neutralized the toxicity of bacterial supernatants (Figure 4B). We also found that the toxicity of supernatant from the ∆phzA1/B1 strain was lower than that from the parent PAO1 strain and the addition of AST-120 to the mutant culture did not further decrease the toxicity to Caco-2 cells (Figure 4C). We assessed the neutralizing effect of AST-120 on cytotoxicity in low doses. We found that treatment with at least 10 mg AST-120 markedly reduced the toxicity of PAO1 culture supernatant to Caco-2 cells (Figure 4D). These results indicate that AST-120 decreases pyocyanin-associated cytotoxicity of *P. aeruginosa* to Caco-2 cells.

### 2.4. AST-120 Does Not Impair Antibacterial Activities of Fosfomycin, Colistin and Amikacin

As an unfavorable property of AST-120, it adsorbs some classes of antimicrobial agents including β-lactams and quinolones. However, fosfomycin, colistin and aminoglycosides are not adsorbed by AST-120 (20). Therefore, fosfomycin, colistin and aminoglycosides, such as amikacin, might still be able to be used together with AST-120 to treat *P. aeruginosa* infections. We estimated the in vitro effectiveness of the combination treatment that used AST-120 and either of these drugs. At the beginning of the experiment, we determined that minimum inhibitory concentrations (MICs) of fosfomycin, colistin and amikacin in PAO1 were 64, 16 and 2 μg/mL, respectively. The PAO1 strains were incubated with either 128 μg/mL fosfomycin, 32 μg/mL colistin or 4 μg/mL amikacin in the presence and absence of 30 mg AST-120. Addition of these drugs inhibited the bacterial growth even when AST-120 was present, suggesting that AST-120 can be used together with fosfomycin, colistin and amikacin (Figure 5).

## 3. Discussion

AST-120 is used to treat progressive CKD. It decreases the production of indoxyl sulfate, a uremic toxin derived from indole produced by some kinds of enteric bacteria [19]. In addition to its role in CKD treatment, we recently found that AST-120 decreased drug tolerance and virulence of *E. coli*, including enterohaemorrhagic *E. coli* [20]. Here, we propose another potential benefit of AST-120 in reducing the cytotoxicity of *P. aeruginosa* by adsorbing phenazines including pyocyanin. 

*P. aeruginosa* is one of the major pathogens that cause nosocomial infections. In addition to the airway, the intestinal tract is considered the primary reservoir of this bacteria in hospitalized patients [23]. *P. aeruginosa* colonized in the gut of immunocompromised patients disrupts the intestinal epithelial barrier and enters the bloodstream, leading to systemic infections. Pyocyanin impairs colon epithelial cells, which may be associated with disruption of the intestinal epithelial barrier. Our results indicated that AST-120 alleviated this effect of pyocyanin. In addition to the cytotoxicity to host epithelial cells, pyocyanin exhibits bactericidal activity against many bacterial species [21,24]. The gut microbiota acts as an intestinal barrier to protect against intestinal pathogens. A recent in vivo study using a mouse model suggested that *E. coli* protected mice from intestinal *P. aeruginosa* [25]. Therefore, adsorption of pyocyanin by AST-120 may protect the host from intestinal *P. aeruginosa* by decreasing the direct damage to intestinal epithelial cells and disturbance of gut microbiota, including *E. coli*, contributing to the homeostasis of the intestinal barrier.

Phenazines, including pyocyanin, also contribute to the biological fitness of *P. aeruginosa* and they may enable the bacteria to establish infection, as these molecules are involved in anaerobic survival, iron acquisition and biofilm development [21,26,27]. Phenazines have also been shown to be involved in the progression of chronic respiratory infections, such as in CF patients [28]. Availability of iron and oxygen for *P. aeruginosa* could be limited in the gut due to the presence of other microorganism competitors, such as gut microbiota. Therefore, phenazines may also aid *P. aeruginosa* to colonize and survive in the gut. AST-120 has the potential to remove the benefits of intestinal *P. aeruginosa* conferred by phenazines. 

Pyocyanin is considered as a potential target for treating *P. aeruginosa* infections. Some methods to alleviate the burden of pyocyanin-associated virulence have been previously proposed. Antioxidant agents have been shown to decrease pyocyanin-induced oxidative stress and improve CF lung function [29]. Pyocyanin is synthesized from shikimic acid via chorismic acid, the phenazine precursor. A pathway to produce chorismic acid from the shikimic acid substrate, termed the *aro* pathway, is found in prokaryotes, but is not present in vertebrates. Thus, *aro* enzymes could be used as attractive therapeutic targets in *P. aeruginosa* [30]. Since the QS systems of *P. aeruginosa* are involved in the induction of pyocyanin biosynthesis, the interference of this system could be a potential alternative therapeutic strategy [22,31]. The adsorption method proposed in this study may be incorporated with novel approaches to prevent and treat *P. aeruginosa* infections.

Although AST-120 adsorbs certain antimicrobial agents [20], it can be used in combination with fosfomycin, colistin and amikacin. Fosfomycin and colistin have recently attracted attention as “last-resort drugs” to treat MDRP strains, which are resistant to commonly used anti-*P. aeruginosa* agents, such as some β-lactams, quinolones and aminoglycosides [32,33]. Therefore, AST-120 may offer a potential benefit by helping chemotherapy treat refractory MDRP infections. In addition to some antimicrobial agents, AST-120 might adsorb some of beneficial secondary metabolites in host cells because it non-selectively adsorbs small-sized molecules containing aromatic ring structures. Several tryptophan metabolites, such as indole propionic acid and indole-3-carboxaldehyde, contribute to the protection of gut epithelial cells [34,35,36]. AST-120 may adsorb these compounds. AST-120 is widely used to treat CKD and it has been approved to be safe even when administrated for a long period. Therefore, non-selective adsorption of AST-120 may be not a critical issue as shown in several clinical trials [37]. However, to enhance the potential efficacy of AST-120 therapy, it is important to establish a protocol that minimizes the interference in the beneficial effects of some secondary metabolites.

## 4. Materials and Methods

### 4.1. Bacterial Strains, Host Cells and Culture Conditions

We used *P. aeruginosa* PAO1 strain, its derivative and some clinical antibiotic-resistant isolates originally isolated from sputum (Ps.a-1200, Ps.a-1205 and RPs.a-914) and urinary catheters (RPs.a-946) in hospitalized patients. The strains are resistant to the antibiotics as follows: Ps.a-1200 (levofloxacin), Ps.a-1205 (aztreonam, meropenem and levofloxacin), RPs.a-946 (imipenem, meropenem, gentamycin, amikacin and levofloxacin) and Rps.a-914 (imipenem, meropenem, gentamycin, amikacin and levofloxacin). The bacteria were aerobically grown in glass tubes with shaking at 160 rpm. Antibiotics were added to the growth media at the following concentrations for marker selection and plasmid maintenance: 300 μg/mL carbenicillin and 100 μg/mL gentamicin for PAO1, 150 μg/mL ampicillin and 20 μg/mL gentamicin for *E. coli*. The A-549 (ATCC CCL-185) and Caco-2 (ATCC HTB-37) cells were obtained from American Type Culture Collection (ATCC). These cells were cultured in Dulbecco’s modified Eagle medium (DMEM) containing 10% HyClone FetalClone III serum (HyClone Laboratories, Inc., Logan, UT, United States) at 37 °C and in an atmosphere of 5% CO_2_.

### 4.2. Mutant Construction

To construct the in-frame deletions of *phzA1* and *phzB1* in the PAO1 background, we performed a gene replacement strategy based on homologous recombination using the pEX18Gm vector as previously described [38,39]. We amplified a flanking DNA fragment including the upstream region of *phzA1* and the downstream region of *phzB1* by sequence overlap extension polymerase chain reaction (PCR) with primer pairs, delta1/delta2 and delta3/delta4 (Table 1). The upstream flanking DNA included 450 bp and the first three amino acid codons. The downstream flanking DNA included the last two amino acid codons, the stop codon and 450 bp of DNA. This deletion construct was ligated into BamHI and HindIII-digested pEX18Gm and introduced into PAO1. We selected sucrose-resistant/gentamicin-sensitive colonies and confirmed the resulting mutant strains using PCR analysis and DNA sequencing.

### 4.3. Pyocyanin Assay

Pyocyanin level was determined as previously described [40], with slight modifications. Bacteria were grown in Luria–Bertani (LB) medium for 24 h and the cell cultures were centrifuged to remove the cell pellets. Pyocyanin in the culture supernatants was extracted with chloroform and then re-extracted into 0.2 M HCl. Pyocyanin was quantified by measuring the absorbance of this solution at 520 nm.

### 4.4. Reporter Construction and Promoter Assays 

To measure the promoter activities of *lasI* and *rhlA* in PAO1, we PCR-amplified the 365- and 425-bp regions upstream of *lasI* and *rhlA* in PAO1, respectively, using the primer pairs listed in Table 1 and ligated each product into the NsiI and KpnI-digested pBBR1MCS4lacZ reporter plasmid [38]. The PAO1 strains carrying each reporter plasmid, designated pBBRlasI (PAO1)-P or pBBRrhlA (PAO1)-P, were grown in the presence and absence of AST-120 and 10% of chloroform was added to lyse the cells. The chemiluminescent signal in the supernatant was generated using a Tropix Galacto-Light Plus kit according to the manufacturer’s instructions (Thermo Fisher Scientific, Waltham, MA, USA). Their β-galactosidase activities corresponding to LacZ expression were determined as the signal value normalized to an OD_600_ of 1 [38].

### 4.5. Cytotoxicity Assays

Bacteria were grown in LB medium for 24 h. The bacteria-free culture supernatants were 2-fold diluted into DMEM containing 10% HyClone FetalClone III serum and added to cultured A-549 and Caco-2 cells in 96-well plates. As a control, a two-fold diluted LB medium in DMEM containing 10% HyClone FetalClone III serum was added to the host cells. After incubation for 24 h, the cell viabilities were determined using CellTiter-Glo Luminescent Cell Viability Assay (Promega Corp., Madison, WI, USA) according to the manufacturer’s instructions. The cell viabilities were represented as relative light units (RLUs) by their ratios (%) to RLU of the control sample.

### 4.6. Minimum Inhibitory Concentration (MIC) Assays

MIC assays were performed by the standard serial agar dilution method of the Clinical and Laboratory Standards Institute (CLSI). The MIC was determined as the lowest concentration at which growth was inhibited.

### 4.7. Statistical Analysis

The *p*-value in each assay was determined by the unpaired t-test with the GraphPad Prism version 6.00.

### 4.8. Approval for Experiments

All experimental protocols were approved by the Gunma University Gene.

Recombination Experiment Safety Committee for all gene recombination and bacteria culture studies (The approval number: 19–041). All experiments were performed in accordance with these committee’s guidelines and regulations.

## Figures and Tables

**Figure 1 antibiotics-10-00434-f001:**
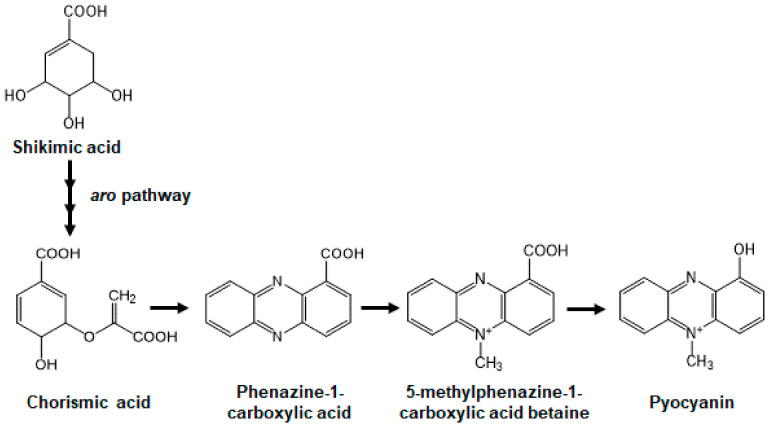
The pathway of pyocyanin production in *P. aeruginosa*.

**Figure 2 antibiotics-10-00434-f002:**
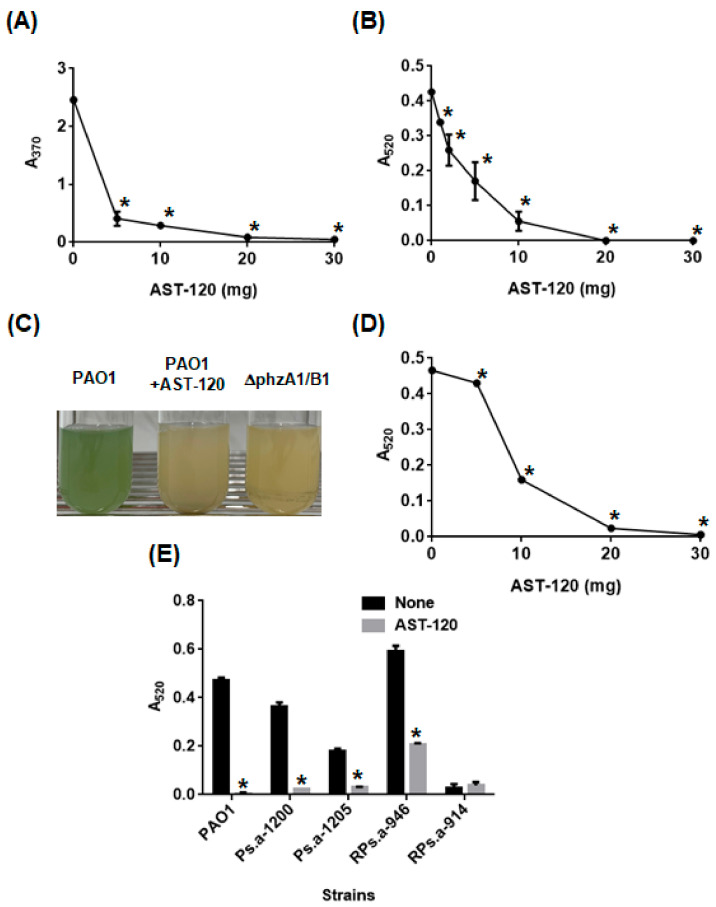
Adsorption and accumulation of phenazine and pyocyanin. (**A**) Phenazine level corresponding to the absorbance of 370 nm in supernatant from an aqueous solution containing 200 μM synthetic phenazine after incubation for 2 h with or without AST-120. (**B**) Pyocyanin level corresponding to the absorbance of 520 nm in supernatant from PAO1 culture after incubation for 2 h with or without AST-120. (**C**) Pyocyanin accumulation in cultures of PAO1 grown with and without 30 mg AST-120 and *phzA1/B1* mutant (designated ∆phzA1/B1). (**D**) Pyocyanin level corresponding to the absorbance of 520 nm in PAO1 culture grown with or without AST-120. (**E**) Pyocyanin level corresponding to the absorbance of 520 nm in PAO1 and *P. aeruginosa* clinical isolates that are resistant to some antimicrobial culture grown with and without 30 mg AST-120. Data plotted are the means from three independent experiments; error bars indicate the standard deviations. Asterisks denote significance for values (*p* < 0.05) of samples or bacterial cultures in the presence of AST-120 relative to those in the absence of AST-120.

**Figure 3 antibiotics-10-00434-f003:**
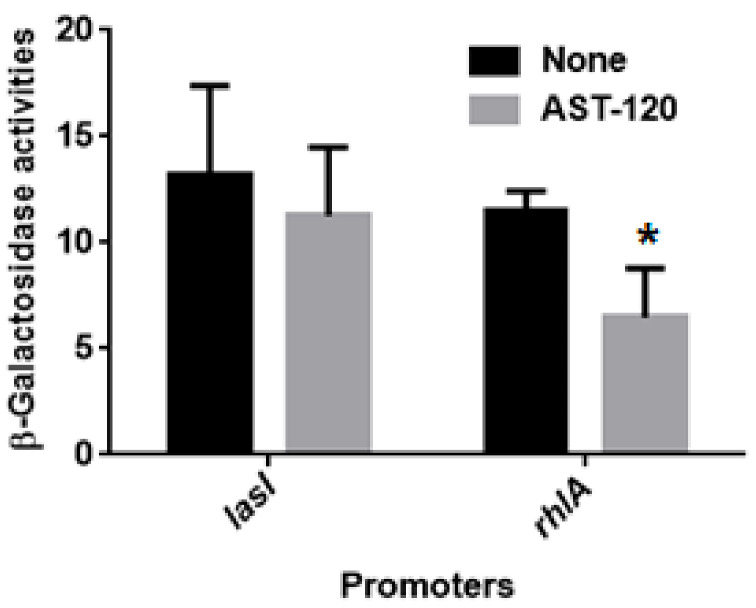
Activities of LasI-LasR and RhlI-RhlR quorum sensing. β-Galactosidase activities of PAO1 containing pBBRlasI (PAO1)-P or pBBRrhlA (PAO1)-P, the *lacZ* reporter plasmid grown with or without 30 mg AST-120, were measured to determine promoter levels of *lasI* and *rhlI* genes regulated by the LasI-LasR and RhlI-RhlR systems. Data plotted are the means from three independent experiments; error bars indicate the standard deviations. Asterisks denote significance for values (*p* < 0.05) of promoter activities in PAO1 grown with AST-120 relative to those in PAO1 grown without AST-120.

**Figure 4 antibiotics-10-00434-f004:**
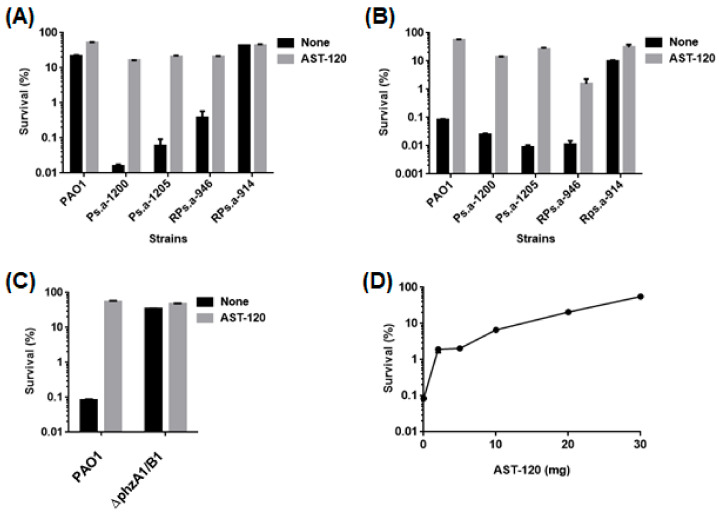
Cytotoxicity of supernatants from PAO1 parent and its *phzA1/B1* mutant strains and drug-resistant clinical isolates. (**A**) Survival of A-549 cells after incubation with or without culture supernatants from indicated *P. aeruginosa* strains cultured in the presence or absence of 30 mg AST-120. (**B**) Survival of Caco-2 cells after incubation with or without culture supernatants from indicated *P. aeruginosa* strains cultured in the presence or absence of 30 mg AST-120. (**C**) Survival of Caco-2 cells after incubation with or without culture supernatants from PAO1 and its *phzA1/B1* mutant (∆phzA1/B1), the phenazines non-producer, cultured in the presence or absence of 30 mg AST-120. (**D**) Survival of Caco-2 cells after incubation with or without culture supernatants from PAO1 cultured in indicated amounts of AST-120. The survival rates are presented as the percentage of the RLU value for the cells after incubation with each supernatant relative to that after incubation without supernatant. Data plotted are the means of two biological replicates, error bars indicate the ranges. We repeated experiments twice, then obtained similar results.

**Figure 5 antibiotics-10-00434-f005:**
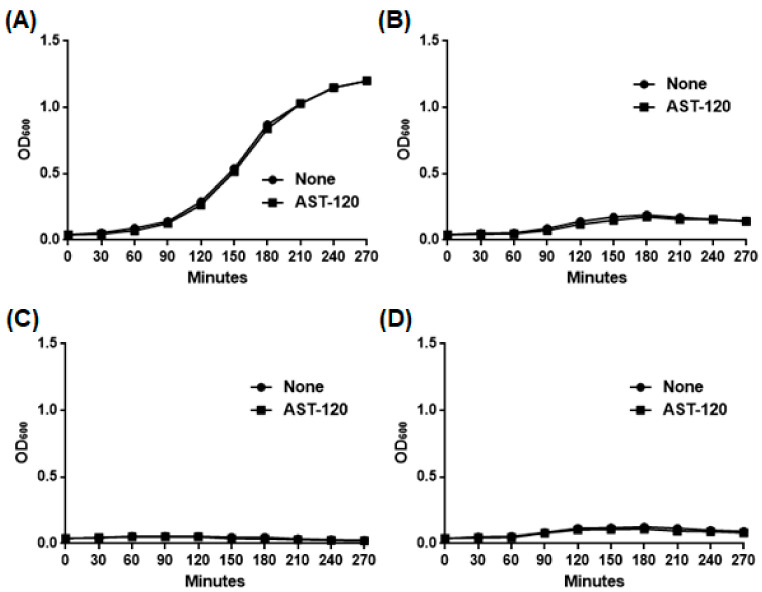
Growth of the PAO1 strain when cultured with or without indicated drugs in the presence and absence of AST-120. Bacteria were grown in LB medium in the presence and absence of 30 mg of AST-120 without drug (**A**) or together with fosfomycin (**B**), colistin (**C**) or amikacin (**D**). The bacterial growth was monitored by measuring OD_600_.

**Table 1 antibiotics-10-00434-t001:** Primers used in this study.

Primer	DNA Sequence (5′–3′)	Use
phzA1B1-delta1 phzA1B1-delta2 phzA1B1-delta3 phzA1B1-delta4 lasI-PF	gcgggatccctacaacctccggcattgc cttccaggatggcctcaggtgggaccgttcatgcgccgcctc ctcggaggcggcgcatgaacggtcccacctgaggccatcctg gcgaagcttcggcgctgttcggcatggg gcgatgcattcttgcctctcaggtcggc	*phzA1* and *phzB1* deletion construction *phzA1* and *phzB1* deletion construction *phzA1* and *phzB1* deletion construction *phzA1* and *phzB1* deletion construction pBBRlasI (PAO1)-P construction
lasI-PR rhlA-PF	gcgggtacccttcacttcctccaaatagg gcgatgcatcgccagagcgtttcgacac	pBBRlasI (PAO1)-P construction pBBRrhlA (PAO1)-P construction
rhlA-PR	gcgggtaccttcacacctcccaaaaattttcg	pBBRrhlA (PAO1)-P construction

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
