# Peer review of "Adsorption of Phenazines Produced by Pseudomonas aeruginosa Using AST-120 Decreases Pyocyanin-Associated Cytotoxicity"

_antibiotics, 2021, doi:10.3390/antibiotics10040434_

Round 1

Reviewer 1 Report

Hirakawa and coauthors report that AST-120 is able to adsorb pyocyanin and its phenazine precursors produced by Pseudomonas aeruginosa. As a result, pyocyanin-associated toxicity in A-549 and Caco-2 cells can be suppressed. It was also demonstrated that AST-120A did not negatively affect the bioactivity of fosfomycin, colistin and amikacin against P. aeruginosa, respectively. Based on these results, authors argued that the benefit of AST-120 may propose a novel method to treat and prevent P. aeruginosa infection. It is well known that AST-120 is an excellent adsorbent that can adsorb many organic molecules. The selective adsorption is a problem. So I am wondering how the adsorption selectivity and efficiency is. For example, does AST-120 will adsorb other normal and useful signal molecules? In this experiment, 30 mg of AST-120A was used. I think the loading is too much, it is not suitable for practical application. As similar studies for the adsorption of AST-120 have been reported (see references 19, 20), this study lack some new ideas or improvement. It cannot be recommended for publication on Antibiotics.

Author Response

Thank you for reviewing our manuscript and providing comments. As you suggested, there is a possibility that AST-120 adsorbs some useful molecules/metabolites in host. For example, some tryptophan metabolites, such as IPA and ICA, that contribute to protection of gut epithelial cells may be adsorbed. However, AST-120 is widely used to treat CKD, and it has been clinically approved to be very safe even in a long-term administration. Therefore, we do not think that the issue on non-selective adsorption of AST-120, is medically critical according to the data in clinical trials. Of course, in case that we use it together with antimicrobial agents to treat infections, we need to consider a possibility that it may adsorb some kinds of antimicrobial agents. However, in this study, we provided an evidence that at least fosfomycin, aminoglycosides and colistin can be used together with AST-120. Especially, fosfomycin and colistin are used to treat MDRP infections (P. aeruginosa strains that are resistant to quinolones, aminoglycosides, and carbapenems). Therefore, AST-120 administration may provide a benefit to improve fosfomycin and colistin therapy to treat MDRP infections. P. aeruginosa is innately highly tolerant to antimicrobial agents, which means a limited chemotherapy option. In addition, strains that developed drug resistance, such as MDRP, are extremely difficult to be treated. We believe that AST-120 offers a potential option to assist anti-Pseudomonas therapy.

Ref19 and 20 showed an ability of AST-120 to adsorb indole and its-related compounds. As far as we know, adsorption of tricyclic compounds containing an aromatic ring, such as phenazines, has not been studied, yet. From these reasons, we believe that this study provides a quite new idea and technology in the field of anti-microbial chemotherapy. We added these statements in Discussion.

6 g of AST-120 per day is administrated to CKD patients. Generally, adsorbent drug is used in higher dose than other typical types of drug, such as receptor antagonists, enzyme inhibitors. Therefore, we think that 30 mg of AST-120 used in this work is not high load. According to several other in vitro studies, more amounts of AST-120 were used (for examples, 500 mg was added in a study [Cancer Chemother Pharmacol. 2007, 59:321-328] and 20 mg/mL was added in a study [Int. J. Mol. Sci. 2019, 20:2280]. In addition, we confirmed that pyocyanin could be adsorbed when even 10 mg AST-120 is present. We had presented the data in Fig. 2. We also tested the effect of AST-120 to neutralize the cytotoxicity of PAO1 supernatant to Caco-2 cells in lower dose (2mg, 5mg, 10mg and 20mg). As well as the experiment of pyocyanin adsorption, 10 mg AST-120 could definitely reduce the cytotoxicity. We provided the additional data in Fig4D.

We would like you to positively reconsider our manuscript for potential publication.

Reviewer 2 Report

The manuscript describes the adsorption potential of AST-120 for pyocyanins and the resulting protection of cells from Pseudomonas aeruginosa toxins.  This manuscript is the next extension of AST-120 antimicrobial applications, which were earlier studied by the authors using Escherichia coli.  

The study is properly designed and well-presented. There are a few minor issues, which should be clarified:

1) Please check and correct numerous English grammar mistakes. The most common issue is the tense inconsistency, for example, in Abstract (lines 16-18) "...AST-120 adsorbs pyocyanin and its phenazine precursors produced by Pseudomonas aeruginosa, including multidrug-resistant P. aeruginosa strains, and suppressed pyocyanin-associated toxicity..." This problem repeatedly appears throughout the manuscript.

2) There are some language style issues. For example, the word "signal" in "Escherichia coli by adsorbing the indole signal" (Abstract, lines 15-16) is redundant and confusing.

3) In Fig.1, please change the label "shikimate" to "shikimic acid" to make it consistent with chorismic acid label, since shikimic acid is drawn in unionized form. 

4) Please remove Fig. 4. It is excessive to illustrate one sentence stating that there was no significant difference between the AST-120-treated and control groups. The same is applicable to Fig. 6. 

5) Lines 188-190: The statement "Since pyocyanin is synthesized from shikimate via phenazine-1-carboxylic acid and 5-methylphenazine-1-carboxylic acid betaine in P. aeruginosa, AST-120 decreases the level of pyocyanin by adsorbing not only pyocyanin itself but also the precursors of pyocyanin production" is too speculative as authors did not study in detail nether concentration of these chemicals nor extent of their adsorption by AST-120.

6) Authors' claims on the ability of AST-120 to "prevent P. aeruginosa infection" or to attenuate its virulence are not substantiated by experimental results and should be removed.

Author Response

The manuscript describes the adsorption potential of AST-120 for pyocyanins and the resulting protection of cells from Pseudomonas aeruginosa toxins.  This manuscript is the next extension of AST-120 antimicrobial applications, which were earlier studied by the authors using Escherichia coli.  

The study is properly designed and well-presented. There are a few minor issues, which should be clarified:

Response: Thank you for reviewing our manuscript and providing comments. We revised our manuscript according to the reviewer’s comments.

1) Please check and correct numerous English grammar mistakes. The most common issue is the tense inconsistency, for example, in Abstract (lines 16-18) "...AST-120 adsorbs pyocyanin and its phenazine precursors produced by Pseudomonas aeruginosa, including multidrug-resistant P. aeruginosa strains, and suppressed pyocyanin-associated toxicity..." This problem repeatedly appears throughout the manuscript.

Response: We carefully checked our English grammar, again and our manuscript received an English correction from English experts served by the Enago.com.

2) There are some language style issues. For example, the word "signal" in "Escherichia coli by adsorbing the indole signal" (Abstract, lines 15-16) is redundant and confusing.

Response: As suggested, we deleted the word of “signal”. Similarly, we also modified Line53-55 to explain more clearly background of indole works as follow. “We previously found that indole induced drug tolerance and virulence in Escherichia coli including enterohaemorrhagic E. coli, and AST-120 abolished these effects by adsorbing indole. Thus, AST-120 reduces drug tolerance and virulence in E. coli.

3) In Fig.1, please change the label "shikimate" to "shikimic acid" to make it consistent with chorismic acid label, since shikimic acid is drawn in unionized form. 

Response: We changed it as suggested. In addition to Fig1, we also changed “shikimate” to “shikimic acid” in the text through the manuscript.

4) Please remove Fig. 4. It is excessive to illustrate one sentence stating that there was no significant difference between the AST-120-treated and control groups. The same is applicable to Fig. 6. 

Response: We removed the figure as suggested, then Fig 5 and Fig 6 in the original version were accordingly renumbered as Fig 4 and Fig 5 in the revised manuscript.

5) Lines 188-190: The statement "Since pyocyanin is synthesized from shikimate via phenazine-1-carboxylic acid and 5-methylphenazine-1-carboxylic acid betaine in P. aeruginosa, AST-120 decreases the level of pyocyanin by adsorbing not only pyocyanin itself but also the precursors of pyocyanin production" is too speculative as authors did not study in detail nether concentration of these chemicals nor extent of their adsorption by AST-120.

Response: We deleted this sentence.

6) Authors' claims on the ability of AST-120 to "prevent P. aeruginosa infection" or to attenuate its virulence are not substantiated by experimental results and should be removed.

Response: We guess that the reviewer indicates our claims described in the Line21-23 and Line56-59 and Line186-187. In Line21-23, we deleted “prevent”. In Line56-59, we changed “attenuates virulence associated with toxicity to host epithelial cells” to “reduces pyocyanin-associated toxicity to host epithelial cells”. In Line186-187, we changed “to reduce the virulence of P. aeruginosa” to “reduce the cytotoxicity of P. aeruginosa”.

Round 2

Reviewer 1 Report

This reviewer appreciates the efforts from the authors to further improve the quality of this manuscript. The main question has been answered. Therefore, it would be suitable for publication on antibiotics.

The English writing should be further improved. Some words are confused.

e.g.

Page 1, line 39: the words “This molecule is synthesized from shikimic acid via chorismic acid and phenazine-1-carboxylic acid” is confused. The presentation need to be improved.

Page 2, line 67: the words “Pyocyanin is a terminal product synthesized from shikimic acid via phenazine compounds…” is confused. The presentation need to be improved.

A suggest for the next study: I would like to see the authors will look deep into the potential negative effect of the adsorption selectivity of AST-120, and how to avoid the negative effect. It is important to do that ahead of the application of AST-12.

Author Response

We deeply appreciate your re-consideration and time. We asked another English editing service (Eigoexpert.com). We believe that our English has been just improved.

e.g.

Page 1, line 39: the words “This molecule is synthesized from shikimic acid via chorismic acid and phenazine-1-carboxylic acid” is confused. The presentation need to be improved.

Response: We rephrased it to simply “This molecule is synthesized from the shikimic acid substrate”.

Page 2, line 67: the words “Pyocyanin is a terminal product synthesized from shikimic acid via phenazine compounds…” is confused. The presentation need to be improved.

Response: We rephrased it to “ Pyocyanin is one of phenazine molecules, and it is produced from its phenazine precursors, such as phenazine-1-carboxylic acid and 5-methylphenazine-1-carboxylic acid betaine”.

A suggest for the next study: I would like to see the authors will look deep into the potential negative effect of the adsorption selectivity of AST-120, and how to avoid the negative effect. It is important to do that ahead of the application of AST-12.

Response: We agree that it is very good point. We will work on it in our future study.